# Symmetry-Aware Fully-Amortized Optimization with Scale Equivariant Graph Metanetworks

**Bart Kuipers**[*]
Institute of Physics
University of Amsterdam

**Freek Byrman**[*]
Informatics Institute
University of Amsterdam

**Daniel Uyterlinde**[*]
Informatics Institute
University of Amsterdam

**Alejandro García-Castellanos**
Amsterdam Machine Learning Lab (AMLab)
University of Amsterdam

## Abstract

Amortized optimization accelerates the solution of related optimization problems by learning mappings that exploit shared structure across problem instances. We explore the use of Scale Equivariant Graph Metanetworks (ScaleGMNs) for this purpose. By operating directly in weight space, ScaleGMNs enable single-shot fine-tuning of existing models, reducing the need for iterative optimization. We demonstrate the effectiveness of this approach empirically and provide a theoretical result: the gauge freedom induced by scaling symmetries is strictly smaller in convolutional neural networks than in multi-layer perceptrons. This insight helps explain the performance differences observed between architectures in both our work and that of Kalogeropoulos et al. (2024). Overall, our findings underscore the potential of symmetry-aware metanetworks as a powerful approach for efficient and generalizable neural network optimization. Open-source code `https://github.com/daniuyter/scalegmn_amortization`.

## 1 Introduction and Background

The exponential growth in neural network (NN) scale necessitates efficient optimization paradigms beyond traditional iterative methods. *Amortized optimization* addresses this challenge by learning to solve families of related optimization problems through shared structural patterns, enabling either direct prediction (fully-amortized) or guided iterative procedures (semi-amortized) [1].

*Metanetworks*, i.e., NNs that operate on other NNs, emerge as a natural framework for amortized optimization. Formally, we consider metanetworks $\hat{f} : \mathcal{G} \times \mathbf{\Theta} \to \mathcal{Y}$ mapping from architecture space $\mathcal{G}$ and parameter space $\mathbf{\Theta}$ to outputs $\mathcal{Y}$. The output can be a scalar, in which case the metanetwork acts as a functional, or a transformed set of parameters, where it acts as an operator.

NNs exhibit *gauge symmetries*, a concept fundamental in physics, originating from non-unique potentials in classical electromagnetism [9]. A gauge symmetry represents invariance where specific transformations of internal parameters leave physical observables unchanged, reflecting redundancies in the system's representation. In NNs, this describes parameter redundancy where different internal configurations produce functionally identical models [6, 18].

Formally, NN symmetries are induced by transformations $\psi : \mathcal{G} \times \mathbf{\Theta} \to \mathcal{G} \times \mathbf{\Theta}$ that preserve network function: $u_{G,\boldsymbol{\theta}}(\mathbf{x}) = u_{\psi(G,\boldsymbol{\theta})}(\mathbf{x}), \forall \mathbf{x} \in \mathcal{X}, \forall (G, \boldsymbol{\theta}) \in \mathcal{G} \times \mathbf{\Theta}$ [26]. Principal symmetries include

---

[*]Equal contribution.

permutation (node reordering within layers) and scaling (weight multiplication/division), creating substantial gauge freedom in parameter selection.

The recognition of these symmetries enables *symmetry-aware metanetworks* that respect gauge invariances by construction rather than learning them through training. This design principle ensures consistency across functionally equivalent representations and enhances learning efficiency by eliminating redundant parameter exploration. Metanetworks acting as functionals must remain *invariant* ($\hat{f}(\psi(G, \boldsymbol{\theta})) = \hat{f}(G, \boldsymbol{\theta})$), while those acting as operators should be *equivariant* ($\hat{f}(\psi(G, \boldsymbol{\theta})) = \psi(\hat{f}(G, \boldsymbol{\theta}))$). Recent advances leverage permutation symmetries via Graph Meta Networks (GMNs) [16, 17], with Scale Equivariant GMNs extending to capture scaling symmetries [10].

**Related Work and Positioning**. An emerging paradigm in metanetwork research is the "learning to optimize" (L2O) framework [2, 1], which trains NNs to serve as optimizers for other networks' parameters. This approach aims to surpass conventional hand-crafted optimization algorithms (SGD, Adam [11]) through learned optimization strategies that can exploit network structure [13]. However, these methods remain inherently iterative, requiring multiple forward passes and objective evaluations to approximate the trajectories of traditional optimizers.

Complementary work on weight prediction [12, 23] generates parameters from computational graphs alone ($\mathcal{G} \rightarrow \boldsymbol{\Theta}$), as initializing weights from scratch requires no knowledge of existing parameters. This approach enables rapid model instantiation but cannot leverage prior optimization states or perform fine-tuning operations.

We propose a fundamentally different paradigm: *single-shot parameter optimization* via $\mathcal{G} \times \boldsymbol{\Theta} \rightarrow \boldsymbol{\Theta}$ mappings that directly transform existing parameters to optimized states, bypassing iterative procedures entirely. Our approach leverages both architectural structure and weight-space symmetries, enabling tasks such as single-shot fine-tuning for improved accuracy or sparsity regularization capabilities that remain underexplored in metanetwork amortization.

This work makes two key contributions to symmetry-aware amortized optimization:

1. Demonstration that ScaleGMNs can be used for effective single-shot optimization across diverse loss landscapes, establishing a new paradigm for efficient NN optimization.

2. Theoretical analysis proving that scaling symmetry gauge freedom is strictly smaller for a CNN layer than for an MLP layer, explaining empirically observed performance differences.

## 2 Methodology

**Scale Equivariant Graph Metanetworks**. The ScaleGMN framework achieves permutation invariance by converting NNs into their graph representations. In this mapping, originally designed by Kofinas et al. [13], weights are mapped to edge features, while biases are mapped to vertex features. A detailed explanation of this mapping, along with a visualization of its translation to CNNs, is provided in the theoretical background section (Appendix, Section B.1) of this work.

Once a NN architecture has been converted into its graph representation, a GMN can operate on its vertex and edge features to construct a metanetwork. The forward pass of a GMN consists of four key stages: feature initialization, message passing, feature updating, and readout. The novel contribution of Kalogeropoulos et al. [10] lies in designing the first three stages to be equivariant and the readout to be invariant to the desired scaling symmetries, thereby yielding an equivariant operator or an invariant functional. This is accomplished by constructing an arsenal of expressive, learnable, equivariant, and invariant helper functions. At the core of this method is the canonical representation of objects. Details on the construction of such a ScaleGMN are provided in Appendix B.3 and B.4.

**Equivariant Amortized Meta-Optimization**. We introduce an amortized meta-optimization model using the ScaleGMN framework. This model is an operator, denoted $\hat{f}_{\boldsymbol{\phi}} : \mathcal{G} \times \boldsymbol{\Theta} \rightarrow \boldsymbol{\Theta}$, parameterized by a set $\boldsymbol{\phi}$, that learns to approximate the solution of the following optimization problem:

$$\hat{f}_{\boldsymbol{\phi}}(G, \boldsymbol{\theta}) \approx f^*(G, \boldsymbol{\theta}) \in \text{argmin}_{\boldsymbol{\theta}' \in \boldsymbol{\Theta}} \mathcal{C}(\boldsymbol{\theta}' \mid \boldsymbol{\theta}, G, \mathcal{D}),$$

where $\mathcal{C}$ is a cost function of a target network $u_{G, \boldsymbol{\theta}'}$ that takes as context the dataset $\mathcal{D}$, the original weights $\boldsymbol{\theta}$, and architecture $G$. For further details on amortized optimization, see Amos [1].

This metanetwork is trained on a collection of networks that are trained on a consistent objective (e.g., classifying CIFAR-10). After training, the operator takes an unseen parameter set $\boldsymbol{\theta}$ from any network with architecture $G$ and directly outputs a new set of parameters $\boldsymbol{\theta}'$, according to the mapping $\hat{f}_\phi(G, \boldsymbol{\theta}) = \boldsymbol{\theta}'$. As illustrated in Figure 1, this process can be visualized in weight space, where the metanetwork maps the network and its parameters to a region of lower cost, thereby guiding the model toward an optimal solution in a *single forward pass*. In the most general optimization setting explored in this work, the ScaleGMN $\hat{f}_\phi$ is trained according to the following objective function:

$$\mathcal{L}(\phi; \boldsymbol{\theta}, \mathcal{B}) = \lambda \cdot \left\| \hat{f}_\phi(G, \boldsymbol{\theta}) \right\|_1 + \frac{1}{|\mathcal{B}|} \sum_{(\mathbf{x}, y) \in \mathcal{B}} \mathcal{L}_{\text{CE}} \left( u_{G, \hat{f}_\phi(G, \boldsymbol{\theta})}(\mathbf{x}), y \right), \quad (1)$$

where the first term represents the $L_1$ regularization on the output parameters, scaled by a hyperparameter $\lambda > 0$, which ensures that the ScaleGMN produces sparse weights. The second term corresponds to the CE loss evaluated over the batch $\mathcal{B}$, applied to the output of the transformed network. This term ensures that the transformed network maintains its classification performance. Note that while the CE loss is typically optimized with respect to the network parameters $\boldsymbol{\theta}$, in this context, we are optimizing over the metanetwork parameters $\phi$. Leveraging the chain rule we find: $\nabla_\phi \mathcal{L}(\phi; \boldsymbol{\theta}, \mathcal{B}) = \nabla_{\boldsymbol{\theta}'} \mathcal{L}(\phi; \boldsymbol{\theta}, \mathcal{B}) \cdot \nabla_\phi \hat{f}_\phi(G, \boldsymbol{\theta})$. The rationale behind the architectural design choices and hyperparameter settings for constructing $\hat{f}_\phi$ is detailed in Appendix D.1.

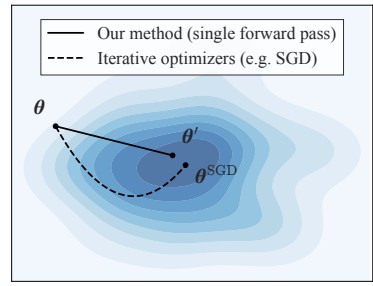

Figure 1: Conceptual idea of our fully-amortized meta-optimizer for a low-dimensional cost function $\mathcal{C}$.

**Experimental Setup**. We validate our method through two primary experiments. The first assesses the core capability of ScaleGMNs for fully-amortized optimization. The second, a symmetry-breaking experiment, is designed to empirically investigate the impact of scale equivariance in this context.

In the amortized optimization experiment, we evaluate our metanetwork's ability to perform a single-shot fine-tuning mapping across diverse loss landscapes defined by different network architectures (CNNs and MLPs), activation functions (Tanh and ReLU), and optimization objectives (standard cross-entropy (CE) and $L_1$-regularized CE). This comprehensive evaluation aims to demonstrate the robustness and generalizability of our fully-amortized optimization approach.

To assess the benefits of symmetry equivariance, we train additional metanetworks for all amortization experiments in which equivariance is deliberately broken by omitting the canonicalization step. This modification preserves the architecture, including parameter count and connectivity, but removes equivariance. We then plot validation loss curves to evaluate the practical impact of scale equivariance.

**Datasets and Baselines**. Our experiments leverage two collections of pretrained models. For CNNs, we use the *Small CNN Zoo dataset* [21]. For MLPs, since no comparable public dataset exists, we constructed our own *Small MLP Zoo*. Both zoos consist of networks trained on CIFAR-10-GS [14], with full generation details provided in Appendix C. As fine-tuning networks via a metanetwork (rather than predicting parameters from scratch) remains underexplored, we do not include other meta fine-tuning methods as baselines. Instead, we establish a baseline using standard SGD-based fine-tuning, similar to the approach of Knyazev et al. [12], who perform parameter prediction from scratch with a metanetwork. Hyperparameter details for this baseline are given in Appendix D.2.

## 3 Results & Discussion

**Equivariant Amortized Meta-Optimization**. The results of the amortization experiments are presented in Table 1. With a single forward pass on both unseen networks and data, it outperforms an SGD baseline trained for over 150 epochs on all CNN tasks and in regularizing MLPs. On MLP cross-entropy optimization, its performance is more modest, equivalent to 25 SGD epochs, a result we attribute to the smaller MLP training set hindering generalization. What makes this result particularly compelling is that our meta-optimizer, which has no prior exposure to the test networks, outperforms an optimizer tailored to each specific instance. While the iterative baseline would likely continue to improve with more epochs, its purpose here is not to be a direct competitor to be surpassed indefinitely,

Table 1: **Evaluation of Amortized Meta-Optimization**. Comparison of CE and $L_1$-regularized experiments on CIFAR-10-GS against iterative baselines for CNN and MLP architectures. Sparsity is defined as the percentage reduction in the total $L_1$ norm of the parameters after fine-tuning. Metanetwork timing was measured as the average forward pass duration over the test set, using a batch size of one. Baseline timing was determined by extrapolating the average per-epoch training time from an initial 30-epoch run, a method found to be consistent across different initializations and activation functions. All measurements were performed on a single NVIDIA A100 GPU.

| Method | Cross-entropy ($\lambda = 0$) | | | L1-regularized ($\lambda > 0$) | | | |
|---|---|---|---|---|---|---|---|
| | Avg Acc (%) | Max Acc (%) | Time (s) | Avg Acc (%) | Test Loss (CE+L1) | Time (s) | Sparsity (%) |
| **CNN Architectures (Tanh)** | | | | | | | |
| *Initial performance* | 37.9 | 48.6 | - | 37.9 | 3.233 | - | 0 |
| *Metanetworks* | | | | | | | |
| ScaleGMN-B | 50.3 | 55.1 | 0.055 | 37.4 | 1.901 | 0.055 | 87.6 |
| GMN-B (sym. broken)[†] | 51.5 | 56.1 | 0.054 | 35.3 | 1.982 | 0.054 | 85.3 |
| *Iterative optimizer* | | | | | | | |
| SGD (25 epochs) | 41.0 | 50.6 | 59.5 | 39.8 | 2.651 | 82.5 | 35.9 |
| SGD (50 epochs) | 41.6 | 50.6 | 119 | 37.9 | 2.474 | 165 | 50.7 |
| SGD (100 epochs) | 43.5 | 51.1 | 238 | 40.4 | 2.189 | 330 | 64.4 |
| SGD (150 epochs) | 44.5 | 51.4 | 357 | 41.6 | 2.070 | 495 | 70.9 |
| **MLP Architectures (Tanh)** | | | | | | | |
| *Initial performance* | 34.5 | 40.2 | - | 34.5 | 3.2581 | - | 0 |
| *Metanetworks* | | | | | | | |
| ScaleGMN-B | 35.4 | 40.7 | 0.056 | 32.7 | 1.964 | 0.070 | 85.2 |
| GMN-B (sym. broken)[†] | 35.0 | 40.4 | 0.054 | 29.8 | 2.074 | 0.068 | 77.4 |
| *Iterative optimizer* | | | | | | | |
| SGD (25 epochs) | 35.5 | 41.5 | 80.0 | 36.6 | 2.180 | 101 | 26.2 |
| SGD (50 epochs) | 35.8 | 41.4 | 160 | 37.1 | 2.095 | 201 | 38.8 |
| SGD (100 epochs) | 36.3 | 41.8 | 320 | 38.0 | 2.016 | 402 | 50.1 |
| SGD (150 epochs) | 36.7 | 41.9 | 480 | 38.6 | 1.963 | 603 | 55.2 |

[†] Identical to ScaleGMN-B, but without the canonicalization step that ensures scale equivariance.

but rather to serve as a benchmark to contextualize the performance of single-shot fine-tuning. For a more detailed distributional analysis and results for ReLU activations, see Appendix E.

**Breaking Scaling Symmetries**. Figures 2a and 2b show that, in the MLP experiments, breaking equivariance leads to a less efficient training trajectory compared to the equivariant model. For CNNs (Figures 2c and 2d), this effect is less pronounced: the loss trajectories of the equivariant and symmetry-broken models are largely similar, with the primary distinction being that the regularized symmetry-broken run fails to reach completion due to numerical instabilities, limiting the conclusiveness of final test accuracy comparisons. Nevertheless, we believe this setup remains the most appropriate for isolating the effect of scaling symmetries, since all other network components are held constant and the training trajectories up to the instability point still provide valuable insights into the underlying dynamics.

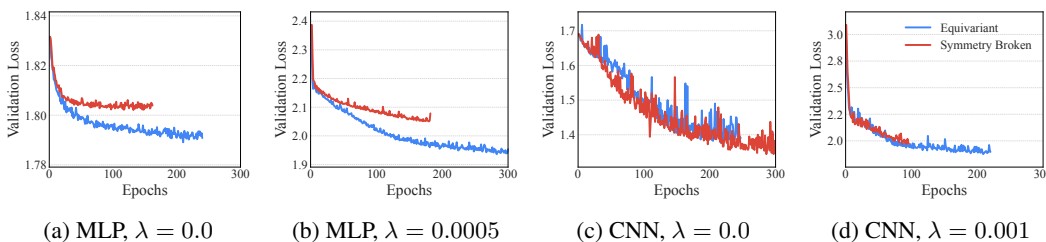

| (a) MLP, $\lambda = 0.0$ | (b) MLP, $\lambda = 0.0005$ | (c) CNN, $\lambda = 0.0$ | (d) CNN, $\lambda = 0.001$ |

Figure 2: **Training Dynamics**. Validation loss curves comparing scale equivariant and symmetry-broken models across different architectures and $L_1$ regularization strengths (w/ tanh activation).

We conclude that the benefits of scale equivariance in the ScaleGMN are more pronounced when processing MLPs compared to CNN architectures. In Kalogeropoulos et al. [10] we find a similar

pattern: the largest gains appear on MLP INR experiments, while CNN generalization prediction improvements are comparatively modest. We hypothesize that this disparity arises from the dimensionality of the gauge group generated by scaling symmetries in CNNs compared to MLPs. Since the effectiveness of ScaleGMN relies on leveraging these symmetries, its potential benefits diminish when the gauge group is small. In Appendix A, we discuss this disparity and formalize it by proving that the gauge group of a CNN layer is a subset of that of an MLP layer with the same I/O dimensions.

## 4 Conclusion and Future Work

This study demonstrates the efficacy of the ScaleGMN framework for single-shot optimization. Our findings reveal that incorporating scale symmetries yields more pronounced benefits for MLPs compared to CNNs, a disparity we attribute to their fundamental differences in gauge freedom. We substantiate this claim with a formal proof that the scaling gauge group of a CNN is a strict subset of an MLP's, a theoretical result likely to generalize across other applications of the framework.

Looking ahead, key avenues for future research include extending this framework to arbitrary network architectures, which was deemed theoretically feasible, as shown in Section 2, but not pursued here due to data and scope limitations. Furthermore, improving the training stability of positive, scale-symmetric models remains a significant open challenge that merits dedicated investigation.

## Acknowledgements

We wish to thank Erik J. Bekkers and Efstratios Gavves, as well as the University of Amsterdam, for providing access to the computational resources for this project. Alejandro García Castellanos is funded by the Hybrid Intelligence Center, a 10-year programme funded through the research programme Gravitation which is (partly) financed by the Dutch Research Council (NWO).

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

# A  Gauge-Symmetry Disparity of CNNs and MLPs

A metanetwork should be robust to functionally equivalent representations of input networks. These equivalences arise from symmetries, such as permutations and scalings of neuron activations, which leave the input network's overall function unchanged.

The value of a metanetwork that is equivariant to these symmetries hinges directly on the extent of the gauge freedom (the set of transformations that preserve a network's function): A larger gauge group corresponds to a greater number of redundant representations of the same function, which an equivariant model can efficiently handle without a corresponding increase in complexity.

In this section, we prove that the gauge freedom associated with scaling symmetries for a convolutional layer is strictly smaller than for a MLP layer with the same input and output dimensions. This highlights a fundamental difference in the representational capacities of CNNs and MLPs and provides a theoretical explanation for why the performance of ScaleGMN on CNN architecture inputs consistently underperformed relative to its performance on MLPs, both in Kalogeropoulos et al. [10] and in this work.

We will do so by first identifying the scaling and permutation symmetries of a general matrix representation of a neural network layer, following Kalogeropoulos et al. [10] (Section A.1). We then demonstrate that both CNN and MLP layers can be expressed in this matrix form, enabling a direct comparative analysis of their symmetries (Section A.2). We continue by analyzing the set of scaling transformations that preserve the structure of each layer's effective weight matrix. We will demonstrate that while any scaling transformation is permissible for the unconstrained weight matrix of an MLP, only a highly restricted subset of transformations can preserve the structured, doubly-block Toeplitz form of the CNN's weight matrix. This will prove that the gauge freedom of the CNN is a strict subset of the MLP's (Section A.3).

## A.1  Neural Network Symmetries

Consider a general neural network layer $\ell$ with input activations represented by the vector $\mathbf{x}_{\ell-1}$ and output activations by the vector $\mathbf{x}_\ell$. The layer's function is determined by its weight matrix $\mathbf{W}_\ell$ and bias vector $\mathbf{b}_\ell$, expressed as: $\mathbf{x}_\ell = \sigma_\ell(\mathbf{W}_\ell \mathbf{x}_{\ell-1} + \mathbf{b}_\ell)$. Now, consider a transformation of these neuron activations by permutation matrices $\mathbf{P}_\ell$ and diagonal scaling matrices $\mathbf{Q}_\ell$ corresponding to layer $\ell$. If the input to layer $\ell$ transforms as: $\mathbf{x}'_{\ell-1} = \mathbf{P}_{\ell-1}\mathbf{Q}_{\ell-1}\mathbf{x}_{\ell-1}$. For the layer to be equivariant, meaning its output transforms correspondingly as $\mathbf{x}'_\ell = \mathbf{P}_\ell\mathbf{Q}_\ell\mathbf{x}_\ell$, the parameters $\mathbf{W}_\ell$ and $\mathbf{b}_\ell$ must transform as follows [10]:

$$\mathbf{W}'_\ell = \mathbf{P}_\ell\mathbf{Q}_\ell\mathbf{W}_\ell\mathbf{Q}_{\ell-1}^{-1}\mathbf{P}_{\ell-1}^{-1}, \qquad \mathbf{b}'_\ell = \mathbf{P}_\ell\mathbf{Q}_\ell\mathbf{b}_\ell \tag{2}$$

Furthermore, this requires the activation function $\sigma_\ell$ to satisfy $\sigma_\ell(\mathbf{P}_\ell\mathbf{Q}_\ell\mathbf{z}) = \mathbf{P}_\ell\mathbf{Q}_\ell\sigma_\ell(\mathbf{z})$, a property common to many element-wise activations under appropriate conditions on $\mathbf{Q}_\ell$ [10]. We can simply verify the transformations in Equation (2) by writing out:

$$
\begin{aligned}
\mathbf{x}'_\ell &= \sigma_\ell(\mathbf{W}'_\ell\mathbf{x}'_{\ell-1} + \mathbf{b}'_\ell) \\
&= \sigma_\ell\left((\mathbf{P}_\ell\mathbf{Q}_\ell\mathbf{W}_\ell\mathbf{Q}_{\ell-1}^{-1}\mathbf{P}_{\ell-1}^{-1})(\mathbf{P}_{\ell-1}\mathbf{Q}_{\ell-1}\mathbf{x}_{\ell-1}) + (\mathbf{P}_\ell\mathbf{Q}_\ell\mathbf{b}_\ell)\right) \\
&= \sigma_\ell\left(\mathbf{P}_\ell\mathbf{Q}_\ell\mathbf{W}_\ell(\mathbf{Q}_{\ell-1}^{-1}\mathbf{P}_{\ell-1}^{-1}\mathbf{P}_{\ell-1}\mathbf{Q}_{\ell-1})\mathbf{x}_{\ell-1} + \mathbf{P}_\ell\mathbf{Q}_\ell\mathbf{b}_\ell\right) \\
&= \sigma_\ell\left(\mathbf{P}_\ell\mathbf{Q}_\ell\mathbf{W}_\ell\mathbf{x}_{\ell-1} + \mathbf{P}_\ell\mathbf{Q}_\ell\mathbf{b}_\ell\right) \qquad\qquad (\text{as } \mathbf{P}_{\ell-1}^{-1}\mathbf{P}_{\ell-1} = \mathbf{I}, \mathbf{Q}_{\ell-1}^{-1}\mathbf{Q}_{\ell-1} = \mathbf{I}) \\
&= \sigma_\ell\left(\mathbf{P}_\ell\mathbf{Q}_\ell(\mathbf{W}_\ell\mathbf{x}_{\ell-1} + \mathbf{b}_\ell)\right) \\
&= \mathbf{P}_\ell\mathbf{Q}_\ell\sigma_\ell(\mathbf{W}_\ell\mathbf{x}_{\ell-1} + \mathbf{b}_\ell) \qquad\qquad (\text{by activation property}) \\
&= \mathbf{P}_\ell\mathbf{Q}_\ell\mathbf{x}_\ell.
\end{aligned}
$$

Indeed, we find that the parameter transformations defined in Equation (2) ensure layer equivariance with respect to neuron activation permutations and scalings. When these transformations are applied consistently across all layers of a network, the internal permutations and scalings effectively cancel out between layers. This implies a gauge freedom in choosing sets of internal parameters, all of which lead to functionally equivalent models. Moreover, the study of these gauge symmetries has been shown to play a crucial role in understanding neural networks' weight space [26, 25, 15, 24, 3].

## A.2 The Matrix Form of CNN and MLP Layers

An MLP layer is inherently structured as a matrix multiplication. The output of a fully connected layer is computed as $\mathbf{x}_\ell = \sigma_\ell(\mathbf{W}_\ell \mathbf{x}_{\ell-1} + \mathbf{b}_\ell)$, where the weight matrix $\mathbf{W}_\ell$ is a dense, unstructured matrix containing all the layer's parameters. This form directly matches the general matrix representation we have defined (Section A.1).

For a CNN, the operation can also be vectorized into this general matrix form. A typical convolutional layer $\ell$ computes the activation $x_{\ell,i}$ at each spatial location $i$ by applying a shared kernel over a local region of the input feature map. The kernel is defined by a set of weights $k_\ell(r)$, indexed by relative positions $r \in \mathcal{R}$, where $\mathcal{R}$ denotes the set of offsets covered by the kernel (e.g., a $3 \times 3$ kernel means $\mathcal{R} = \{-1, 0, +1\}^2$). At each location $i$, the kernel is centered and applied to the corresponding input patch, producing a weighted sum of input activations $x_{\ell-1,i+r}$. A shared bias term $b_\ell$ is added, and the result is passed through a nonlinear activation function $\sigma_\ell$:

$$x_{\ell,i} = \sigma_\ell \left( \sum_{r \in \mathcal{R}} k_\ell(r) \cdot x_{\ell-1,i+r} + b_\ell \right). \tag{3}$$

This operation can be vectorized for the entire layer as: $\mathbf{x}_\ell = \sigma_\ell(\mathbf{W}_\ell \mathbf{x}_{\ell-1} + \mathbf{b}_\ell)$. Here, $\mathbf{x}_{\ell-1}$ and $\mathbf{x}_\ell$ are vectors representing the neuron activations at the input and output of the layer, respectively. The vector $\mathbf{b}_\ell$ is the corresponding effective bias vector, formed by appropriately replicating the bias term associated with each filter across all spatial locations where the filter is applied. The matrix $\mathbf{W}_\ell$ is an effective weight matrix that represents the linear transformation performed by the convolution. Due to the shared filter weights and local receptive fields inherent in CNNs, $\mathbf{W}_\ell$ is a sparse matrix with a structured pattern. For the 2D convolutions, this is a *doubly-block Toeplitz matrix* [5]. A doubly-block Toeplitz matrix is a highly structured matrix: It's a block Toeplitz matrix where each block is itself a Toeplitz matrix*. Its entries are derived from the shared filter weights, arranged according to the specific connections (receptive fields, stride) defined by the convolution. A simple example of such a weight representation of a convolution is given below:

Consider a $3 \times 3$ input feature map $\mathbf{x}_{\ell-1}$ and a $2 \times 2$ convolutional kernel $\mathbf{K}$, with a stride of 1 and no padding.

Feature map $\mathbf{x}_{\ell-1}$: $\begin{pmatrix} x_1 & x_2 & x_3 \\ x_4 & x_5 & x_6 \\ x_7 & x_8 & x_9 \end{pmatrix}$ , Convolutional kernel $\mathbf{K}$: $\begin{pmatrix} k_1 & k_2 \\ k_3 & k_4 \end{pmatrix}$

We can now verify that a matrix $\mathbf{W}_\ell$ can be constructed, from the kernel weights such that the vectorized convolution is performed as $\mathbf{x}_\ell = \mathbf{W}_\ell \mathbf{x}_{\ell-1}$:

$$\mathbf{W}_\ell \mathbf{x}_{\ell-1} = \begin{pmatrix} k_1 & k_2 & 0 & k_3 & k_4 & 0 & 0 & 0 & 0 \\ 0 & k_1 & k_2 & 0 & k_3 & k_4 & 0 & 0 & 0 \\ 0 & 0 & 0 & k_1 & k_2 & 0 & k_3 & k_4 & 0 \\ 0 & 0 & 0 & 0 & k_1 & k_2 & 0 & k_3 & k_4 \end{pmatrix} \begin{pmatrix} x_1 \\ x_2 \\ x_3 \\ x_4 \\ x_5 \\ x_6 \\ x_7 \\ x_8 \\ x_9 \end{pmatrix} = \begin{pmatrix} k_1 x_1 + k_2 x_2 + k_3 x_4 + k_4 x_5 \\ k_1 x_2 + k_2 x_3 + k_3 x_5 + k_4 x_6 \\ k_1 x_4 + k_2 x_5 + k_3 x_7 + k_4 x_8 \\ k_1 x_5 + k_2 x_6 + k_3 x_8 + k_4 x_9 \end{pmatrix}$$

The elements of the resulting vector $\mathbf{x}_\ell$ correspond to the output of a standard (cross-correlation) convolution. The weight matrix $\mathbf{W}_\ell$ exhibits the typical doubly-block Toeplitz structure, which enforces weight sharing across rows: each kernel weight appears in multiple positions corresponding to different rows. For instance, with stride 1 and padding 0 (as in our example), each row is simply a permutation of the others. Standard convolution architectures are designed to reuse weights across spatial positions. Even with arbitrary but "reasonable" stride and padding, any two rows share at least one kernel weight. Configurations in which rows contain completely disjoint kernel sets would eliminate weight sharing, a design that is generally not intended. Therefore, for all practical

---

*In a *Toeplitz matrix* each descending diagonal from left to right is constant. Formally, an $n \times n$ Toeplitz matrix $\mathbf{A}$ satisfies $A_{i,j} = c_{i-j}$ for some constants $c_{-(n-1)}, \ldots, c_{n-1}$, meaning that each entry depends only on the difference $i - j$.

convolutions, each kernel weight appears in multiple rows. We will exploit this property in the next section.

### A.3 Proof: The Gauge Group of CNNs is a Strict Subset of MLPs

The CNN's weight matrix $\mathbf{W}$ is restricted to the doubly-block Toeplitz form. In what follows, we formally prove that this structural constraint strictly reduces the gauge freedom associated with scaling symmetries compared to a fully connected MLP.

**Lemma 1.** *Consider a neural network layer with n input features and m output features (a single feature map in the CNN case). Let $\mathcal{W}_{\mathrm{MLP}} = \mathbb{R}^{n \times m}$ be the space of weight matrices for a fully-connected layer (MLP), and let $\mathcal{W}_{\mathrm{CNN}} \subset \mathbb{R}^{n \times m}$ be the space of convolutional weight matrices (doubly-block Toeplitz). Then, the set of admissible pairs of diagonal scaling matrices that preserve $\mathcal{W}_{\mathrm{CNN}}$ is a strict subset of those that preserve $\mathcal{W}_{\mathrm{MLP}}$.*

*Formally, define the set of scaling transformations that preserve a weight space $\mathcal{W}$ as*

$$\mathcal{G}_{\mathrm{scale}}(\mathcal{W}) = \left\{ (\mathbf{Q}_\ell, \mathbf{Q}_{\ell-1}) \in \mathrm{Diag}(n) \times \mathrm{Diag}(m) \,\middle|\, \mathbf{Q}_\ell \mathbf{W} \mathbf{Q}_{\ell-1}^{-1} \in \mathcal{W} \text{ for all } \mathbf{W} \in \mathcal{W} \right\}.$$

*Then:*

$$\mathcal{G}_{\mathrm{scale}}(\mathcal{W}_{\mathrm{CNN}}) \subsetneq \mathcal{G}_{\mathrm{scale}}(\mathcal{W}_{\mathrm{MLP}}).$$

*Proof.* For a fully connected MLP layer, the weight matrix $\mathbf{W} \in \mathcal{W}_{\mathrm{MLP}}$ is arbitrary ($\mathcal{W}_{\mathrm{MLP}} = \mathbb{R}^{n \times m}$). For any pair of invertible diagonal matrices $(\mathbf{Q}_\ell, \mathbf{Q}_{\ell-1}) \in \mathrm{Diag}(n) \times \mathrm{Diag}(m)$, the transformed matrix $\mathbf{W}' = \mathbf{Q}_\ell \mathbf{W} \mathbf{Q}_{\ell-1}^{-1}$ also lies in $\mathbb{R}^{n \times m}$. Thus,

$$\mathcal{G}_{\mathrm{scale}}(\mathcal{W}_{\mathrm{MLP}}) = \mathrm{Diag}(n) \times \mathrm{Diag}(m).$$

This gauge group has a dimension of n+m.

Next, consider the CNN case. Let $\mathbf{W} \in \mathcal{W}_{\mathrm{CNN}}$ be a doubly-block Toeplitz matrix. The structure of $\mathbf{W}$ is defined by a small set of shared kernel parameters. Specifically, entries of $\mathbf{W}$ corresponding to the same kernel weight are equal.

Applying a scaling transformation,

$$\mathbf{W}' = \mathbf{Q}_\ell \mathbf{W} \mathbf{Q}_{\ell-1}^{-1}.$$

Because $\mathbf{Q}_\ell$ and $\mathbf{Q}_{\ell-1}$ are diagonal, this results in an element-wise transformation:

$$W'_{ij} = (\mathbf{Q}_\ell)_{ii} \cdot W_{ij} \cdot (\mathbf{Q}_{\ell-1}^{-1})_{jj}.$$

It follows that the $i$-th row of $\mathbf{W}$ is scaled uniformly by $(\mathbf{Q}_\ell)_{ii}$. Now, because $\mathbf{W}_\ell$ is a doubly-block Toeplitz matrix representing a convolution, and under reasonable[†] stride and padding conditions, every row shares at least one non zero entry (kernel weight) with the subsequent row. This shared structure means that scaling any row independently would violate the equality of shared weights across rows. Therefore, to preserve the doubly-block Toeplitz structure, all diagonal elements of $\mathbf{Q}_\ell$ must be equal. By the same argument, since $\mathbf{Q}_{\ell-1}^{-1}$ acts as the corresponding scaling factor for the preceding layer, its diagonal elements must also be identical. Consequently, the scaling matrices take the form

$$\mathbf{Q}_\ell = \alpha \mathbf{I}, \quad \mathbf{Q}_{\ell-1} = \beta \mathbf{I},$$

where $\mathbf{I}$ denotes the identity matrix and $\alpha, \beta \in \mathbb{R}$. Thus,

$$\mathcal{G}_{\mathrm{scale}}(\mathcal{W}_{\mathrm{CNN}}) = \{\alpha \mathbf{I} \mid \alpha \in \mathbb{R}\} \times \{\beta \mathbf{I} \mid \beta \in \mathbb{R}\}$$

The dimension of this gauge group is 2, corresponding to the two independent scaling factors $\alpha$ and $\beta$. However, since a layer's weights are only affected by the ratio of these factors ($\mathbf{W}' = \alpha \mathbf{W} \beta^{-1}$), the effective gauge freedom is one-dimensional per layer. More generally, for a CNN layer with $n$ channels, the effective gauge freedom is $n$ dimensional.

This condition is stricter than what is allowed for an unconstrained matrix and holds only for specific, constrained choices of $\mathbf{Q}_\ell$ and $\mathbf{Q}_{\ell-1}$ (e.g., uniform scaling factors). A general choice of $\mathbf{Q}_\ell$ and $\mathbf{Q}_{\ell-1}$ will destroy the Toeplitz structure. Therefore, the set of admissible scaling transformations is a strict subset. Hence:

$$\mathcal{G}_{\mathrm{scale}}(\mathcal{W}_{\mathrm{CNN}}) \subsetneq \mathcal{G}_{\mathrm{scale}}(\mathcal{W}_{\mathrm{MLP}})$$

$\square$

---

[†]As discussed in Section A.2

This proof is in agreement with the conceptual understanding that the gauge group of convolutional neural networks is smaller due to weight sharing, as noted in recent work by Hashimoto et al. [6]. We aimed to formalize this relationship by showing that the set of admissible scaling transformations for a CNN layer is a strict subset of that for an MLP layer. In practice, a neural network is formed by a series of layers. In this multi-layered context, we find that a CNN's gauge freedom (the dimension of the gauge group) equals the sum of its channels across all layers. For an MLP we find $N_{\text{neurons}} + N_{\text{input}}$ degrees of freedom across the entire network, where $N_{\text{neurons}}$ is the sum of neurons in all hidden layers and $N_{\text{input}}$ is the number of network inputs. This value is typically much larger than the total channel count of a CNN. This difference in the number of symmetries explains why methods like ScaleGMN, which are designed to exploit these scaling symmetries, offer limited benefits when applied to convolutions, as they possess far fewer symmetries to exploit.

### A.3.1 Activation-Specific Gauge Groups

Lemma 1 is stated and proven for the most general set of scaling matrices, $(\mathbf{Q}_\ell, \mathbf{Q}_{\ell-1}) \in \text{Diag}(n) \times \text{Diag}(m)$. In practice, however, the admissible scaling transformations are further constrained by the activation function $\sigma_\ell$. As stated in Appendix A.1, scale equivariance requires

$$\sigma_\ell(\mathbf{P}_\ell \mathbf{Q}_\ell \mathbf{z}) = \mathbf{P}_\ell \mathbf{Q}_\ell \sigma_\ell(\mathbf{z}),$$

which restricts the allowable diagonal matrices $\mathbf{Q}_\ell$. Thus, while the proof of Lemma 1 applies in complete generality, the specific activation function determines the subset of scaling matrices that are valid in practice.

**Examples.**

- **ReLU:** For the Rectified Linear Unit, $\sigma(z) = \max(0, z)$, the condition holds for any diagonal matrix with strictly positive entries. We denote this set as $\text{Diag}^+(k)$, where $k$ is the dimensionality.

- **Tanh:** For the hyperbolic tangent activation, $\sigma(z) = \tanh(z)$, which is an odd function, the condition is satisfied only if the diagonal entries of $\mathbf{Q}_\ell$ are restricted to $\{\pm 1\}$. We denote this set as $\text{Diag}^\pm(k)$.

These activation-specific restrictions reduce the set of admissible scalings but do not alter the general validity of the proof. For the two activations used in this work, the corresponding gauge groups become:

$$\mathcal{G}_{\text{Tanh}}(\mathcal{W}) = \begin{cases} \text{Diag}^\pm(n) \times \text{Diag}^\pm(m), & \mathcal{W} = \mathcal{W}_{\text{MLP}}, \\ \{\alpha\mathbf{I} \mid \alpha \in \{\pm 1\}\} \times \{\beta\mathbf{I} \mid \beta \in \{\pm 1\}\}, & \mathcal{W} = \mathcal{W}_{\text{CNN}}. \end{cases}$$

$$\mathcal{G}_{\text{ReLU}}(\mathcal{W}) = \begin{cases} \text{Diag}^+(n) \times \text{Diag}^+(m), & \mathcal{W} = \mathcal{W}_{\text{MLP}}, \\ \{\alpha\mathbf{I} \mid \alpha \in \mathbb{R}^+\} \times \{\beta\mathbf{I} \mid \beta \in \mathbb{R}^+\}, & \mathcal{W} = \mathcal{W}_{\text{CNN}}. \end{cases}$$

In both cases, the inclusion

$$\mathcal{G}_{\text{scale}}(\mathcal{W}_{\text{CNN}}) \subsetneq \mathcal{G}_{\text{scale}}(\mathcal{W}_{\text{MLP}})$$

remains valid.

## B  Theoretical background: Symmetry preserving graph meta neural nets

This section is included to provide theoretical background and is not part of the novel experiments presented in this work. Where possible, we adopt the notation of the original work [10], while presenting their framework from our perspective.

We begin by explaining how ScaleGMN achieves permutation symmetry, considering the mapping from MLPs and CNNs to their graph representations (Appendix B.1). We then identify how the redundant scaling transformations (Equation 2) appear in this graph structure (B.2). Finally, we discuss how ScaleGMN respects these symmetries by constructing scale-equivariant and invariant functions that operate on the graph representation to acchive its operator (B.3) or functional form (B.4).

## B.1 Permutation Symmetry: Neural Networks as Graphs

The permutation symmetry of the scaleGMN is achieved by mapping the representing NNs to their inherently permutation-invariant graph representations. In this mapping, originally designed by Kofinas et al. [13], weights are mapped to edge features, while biases are mapped to vertex features.

This graph, subsequently processed by a metanetwork, is formally denoted as $G = (\mathcal{V}, \mathcal{E})$, where $\mathcal{V}$ represents the set of vertices and $\mathcal{E}$ the set of edges. Vertex features are denoted by $\mathbf{x}_V \in \mathbb{R}^{|\mathcal{V}| \times d_v}$, and edge features are denoted by $\mathbf{x}_E \in \mathbb{R}^{|\mathcal{E}| \times d_e}$. For an FFNN, constructing vertex and edge features is intuitive. Vertex features are associated with neuron biases, while edge features represent the connection weights between neurons. This typically results in feature dimensions of $d_v = 1$ and $d_e = 1$. In contrast, for a CNN, the mapping is more nuanced. Vertex features remain scalar ($d_e = 1$) and correspond to the biases of convolutional kernels and neurons in fully connected layers. Edge features, however, are structured to capture the convolutional weights. To ensure a uniform representation, each edge feature is padded to a fixed size $d_e = w_{\max} \cdot h_{\max}$, where $w_{\max}$ and $h_{\max}$ denote the maximum kernel width and height across all channels, respectively. Edges corresponding to weights in fully connected layers are also mapped to this unified feature dimension to maintain consistency across the network representation. An example of this mapping is illustrated in Figure 3.

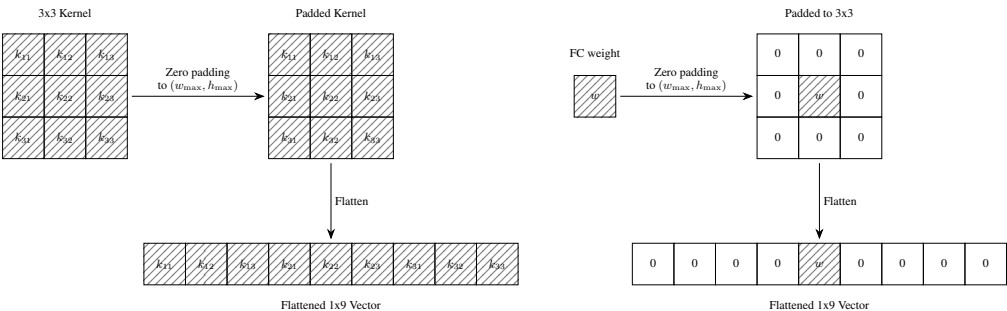

(a) Edge features from a convolutional kernel.       (b) Edge features from a fully connected weight.

Figure 3: Illustration of edge feature representations for different edge types: (a) edges corresponding to convolutional kernels and (b) edges corresponding to weights in fully connected layers, with $w_{\max} = h_{\max} = 3$. This figure is inspired by Figure 6 in Appendix C of Kofinas et al. [13].

Figure 4 illustrates a CNN alongside its corresponding graph representation. The input node corresponds to the number of input channels, 1 in this example, as the network processes grayscale images. Since input vertices do not represent actual parameters, their associated feature vectors are set to zero.

Once a NN architecture has been converted into its graph representation, as illustrated in Figure 4, we can perform operations on the vertex and edge features using a GMN. The forward pass of a GMN with $T$ layers consists of four key stages: feature initialization, message passing, feature updating, and readout:

$$\mathbf{h}_V^0(i) = \text{INIT}_V(\mathbf{x}_V(i)), \quad \mathbf{h}_E^0(i,j) = \text{INIT}_E(\mathbf{x}_E(i,j)) \tag{Init}$$

$$\mathbf{m}_V^t(i) = \bigoplus_{j \in \mathcal{N}(i)} \text{MSG}_V^t\left(\mathbf{h}_V^{t-1}(i), \mathbf{h}_V^{t-1}(j), \mathbf{h}_E^{t-1}(i,j)\right) \tag{Msg}$$

$$\mathbf{h}_V^t(i) = \text{UPD}_V^t(\mathbf{h}_V^{t-1}(i), \mathbf{m}_V^t(i)), \quad \mathbf{h}_E^t(i,j) = \text{UPD}_E^t\left(\mathbf{h}_V^{t-1}(i), \mathbf{h}_V^{t-1}(j), \mathbf{h}_E^{t-1}(i,j)\right) \tag{Upd}$$

$$\mathbf{h}_G = \text{READ}\left(\{\mathbf{h}_V^T(i)\}_{i \in \mathcal{V}}\right), \tag{Readout}$$

where $\mathbf{h}_V^t, \mathbf{h}_E^t$ are vertex and edge representations in the $t^{th}$ layer. The functions INIT, MSG, and UPD are general function approximators. In Kalogeropoulos et al. [10], these functions are designed to be equivariant to scaling transformations. Additionally, the READ function, beyond being permutation invariant, is also required to be invariant to different scalar multipliers applied to each vertex. Further details on these symmetry-preserving functions are provided in Appendix B.3.

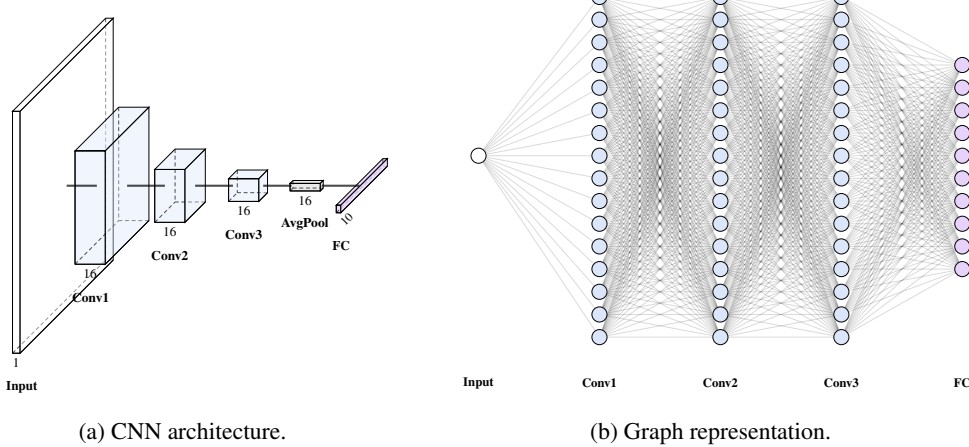

|                          |                          |
|:------------------------:|:------------------------:|
| (a) CNN architecture.    | (b) Graph representation.|

Figure 4: Visualization of the mapping from a CNN to its corresponding graph structure. The CNN illustrated is an example architecture from the Small CNN Zoo [21]. Our methodology follows the approach described in Kofinas et al. [13] for constructing graph representations of CNNs.

## B.2    Mapping CNN Symmetries to Graph Features

A metanetwork designed to process CNNs should be invariant or equivariant to functionally equivalent representations of its input. Because the metanetwork operates on the graph representation of the CNN, we examine how the (functionally equivalent) transformations of Equation (2) transform the corresponding graph representation of the CNN. In this graph, each neuron $i$ from CNN layer $\ell$, with bias $b_{\ell,i}$, is a vertex with feature $\mathbf{h}_V(i) = b_{\ell,i} \in \mathbb{R}$. An edge connecting a "sending" neuron $s$ (CNN layer $\ell - 1$) to a "receiving" neuron $r$ (CNN layer $\ell$), has an edge feature $\mathbf{h}_E(r, s) \in \mathbb{R}^{d_e}$ representing the flattened kernel. Edge features are zero-padded to $d_e = w_{\max} \cdot h_{\max}$ to unify kernel dimensions, as visualized in Figure 3. Consider a symmetry operation at layer $\ell$ defined by a permutation $\pi_\ell : \mathcal{V}_\ell \to \mathcal{V}_\ell$ and a scaling function $q_\ell : \mathcal{V}_\ell \to \mathbb{R} \setminus \{0\}$. The vertex and edge features then transform according to Equation (2) as follows:

$$\mathbf{h}'_V(i) = q_\ell(\pi_\ell(i))\mathbf{h}_V(\pi_\ell(i)), \tag{4}$$

$$\mathbf{h}'_E(r, s) = q_\ell(\pi_\ell(r))\mathbf{h}_E(\pi_\ell(r), \pi_{\ell-1}(s))q_{\ell-1}^{-1}(\pi_{\ell-1}(s)). \tag{5}$$

Next, we will demonstrate how to build a metanetwork (ScaleGMN) that is designed to be invariant/equivariant to these transformations.

## B.3    ScaleEquivariant Net

The novel contribution of Kalogeropoulos et al. [10] is recognizing that applying scale-equivariant initialization, message passing, and update functions to a graph representation of a network (Appendix B.1) results in an equivariant metanetwork. They accomplish this by building an arsenal of expressive, learnable, equivariant, and invariant helper functions. At the heart of this method lies the canonical representation of objects. We define $\tilde{\mathbf{x}} := \mathrm{canon}(\mathbf{x}) : \mathcal{X} \to \mathcal{X}$, where $\mathrm{canon}(\mathbf{x})$ denotes the canonical representative of $\mathbf{x}$ under the equivalence relation $\sim$ induced by scaling transformations. In other words, all elements of $\mathcal{X}$ that are equivalent under scaling are mapped to the same canonical form. Formally, for any $\mathbf{x}, \mathbf{y} \in \mathcal{X}$, $\mathbf{x} \sim \mathbf{y} \implies \tilde{\mathbf{x}} = \tilde{\mathbf{y}}$. In the case of scaling transformations, a natural canonical mapping is given by $\tilde{x} = x/|x|$. This transformation allows for any function $\rho(\mathbf{x})$ to be converted into a scale-invariant function by applying it to the canonicalized input $\rho(\tilde{\mathbf{x}})$. In the ScaleGMN framework [10], a MLP is employed for this base function $\rho$, leveraging its capabilities as a universal approximator. This invariant MLP belonging to layer $k$ is denoted as:

$$\mathrm{ScaleInv}^k(\mathbf{X}) = \rho^k(\tilde{\mathbf{x}}_1, \dots, \tilde{\mathbf{x}}_n). \tag{6}$$

For a scale-equivariant network, the goal is to construct functions $f$ such that if the input $\mathbf{X}$ is scaled by a factor $q$ (i.e., each component $\mathbf{x}_i$ of $\mathbf{X}$ becomes $q\mathbf{x}_i$), the output also scales by the same factor: $f(q\mathbf{X}) = qf(\mathbf{X})$. Linear transformations $\Gamma : \mathcal{X} \to \mathbb{R}^d$ inherently possess this property,

as $\Gamma(q\mathbf{x}) = q(\Gamma\mathbf{x})$. Building on this set of transformations, we can construct more expressive equivariant functions by recognizing that if a scale-equivariant function is multiplied (element-wise $\odot$) by a scale-invariant quantity, the result is itself scale-equivariant [‡]. Kalogeropoulos et al. [10] use this generalization to construct a learnable ScaleEquivariant function, potentially composed of $K$ layers $f^1, \ldots, f^K$, and denote each layer $f^k$ as:

$$\text{ScaleEq} = f^K \circ \cdots \circ f^1, \quad f^k(\mathbf{X}) = (\Gamma_1^k \mathbf{x}_1, \ldots, \Gamma_n^k \mathbf{x}_n) \odot \text{ScaleInv}^k(\mathbf{X}). \tag{7}$$

Here, $\mathbf{X} = (\mathbf{x}_1, \ldots, \mathbf{x}_n)$ is the input to the $k$-th layer. $\Gamma_i^k$ are learnable linear transformations applied to each respective input component $\mathbf{x}_i$. $(\Gamma_1^k \mathbf{x}_1, \ldots, \Gamma_n^k \mathbf{x}_n)$ is a concatenated vector which scales by $q$ if each $\mathbf{x}_i$ scales by $q$. The term $\text{ScaleInv}^k(\mathbf{X})$ is the scale-invariant function defined previously, whose output does not change with the scaling of $\mathbf{X}$. The Hadamard product $\odot$ then ensures that the overall output $f^k(\mathbf{X})$ scales by $q$, thus achieving scale equivariance for the layer.

This general learnable equivariant function (ScaleEq) can now be used to construct equivariant initialization (INIT), message (MSG), and update (UPD) functions. For initialization, Kalogeropoulos et al. [10] use the simplest instance of the equivariant function (Equation 7) with one layer ($k = 1$), and where the invariant component is omitted: $\text{INIT}_{V,E}(\mathbf{x}) = \mathbf{\Gamma}\mathbf{x}$.

Integrating the scale equivalence into the MSG component of the Graph Neural Network is less trivial. The primary challenge arises from the nature of message passing in GNNs, which involves interactions between different types of features that may not share the same scaling behavior. A message function processes features from a sending node, a receiving node, and the connecting edge. If these components $(\mathbf{x}_{\text{sender}}, \mathbf{x}_{\text{receiver}}, \mathbf{e}_{\text{edge}})$ scale by different factors ($q_s, q_r, q_e$ respectively) under a global transformation, the simple ScaleEq formulation (7), is not directly applicable. Ultimately, for a GNN to achieve scale equivariance, the message passed to a receiving node should scale proportionally with that receiving node's scaling factor. This ensures that the information flow remains consistent with the node's transformed representation, e.g the message function should satisfy $\text{MSG}_V(q_r\mathbf{x}_r, q_s\mathbf{x}_s, q_e\mathbf{e}) = q_r\text{MSG}_V(\mathbf{x}_s, \mathbf{x}_r, \mathbf{e})$. To address this Kalogeropoulos et al. [10] first develop a learnable *rescale invariant* function $g$ that is defined by the property: $g(q_1\mathbf{x}_1, \ldots, q_n\mathbf{x}_n) = g(\mathbf{x}_1, \ldots, \mathbf{x}_n)\prod_{i=1}^n q_i, \quad \forall q_i \in D_i$. In the ScaleGMN equivariant framework, $g$ is implemented as a set of component-wise learnable linear transformations followed by an element-wise product aggregation:

$$\text{ReScaleEq}(\mathbf{x}_1, \ldots, \mathbf{x}_n) = \bigodot_{i=1}^n \Gamma_i \mathbf{x}_i. \tag{8}$$

It trivially follows that this function satisfies the conditions for rescale invariance [§]. Leveraging this function, the scale equivariant message update function can now be constructed as:

$$\text{MSG}_V(\mathbf{x}_r, \mathbf{x}_s, \mathbf{e}) = \text{ScaleEq}([\mathbf{x}_r, \text{ReScaleEq}(\mathbf{x}_s, \mathbf{e})]). \tag{9}$$

Indeed, after applying the scaling transformations to vertex and edge features, as defined by Equations (4) and (5) respectively, we can demonstrate the desired equivariance property of the message function $\text{MSG}_V$:

$$\begin{aligned}
\text{MSG}_V(q_r\mathbf{x}_r, q_s\mathbf{x}_s, q_r\mathbf{e}q_s^{-1}) &= \text{ScaleEq}\left([q_r\mathbf{x}_r, \text{ReScaleEq}(q_s\mathbf{x}_s, q_r\mathbf{e}q_s^{-1})]\right) \\
&= \text{ScaleEq}\left([q_r\mathbf{x}_r, (q_s)(q_rq_s^{-1})\text{ReScaleEq}(\mathbf{x}_s, \mathbf{e})]\right) \\
&= \text{ScaleEq}([q_r\mathbf{x}_r, q_r\text{ReScaleEq}(\mathbf{x}_s, \mathbf{e})]) \\
&= q_r\text{ScaleEq}([\mathbf{x}_r, \text{ReScaleEq}(\mathbf{x}_s, \mathbf{e})]) \\
&= q_r\text{MSG}_V(\mathbf{x}_r, \mathbf{x}_s, \mathbf{e}).
\end{aligned}$$

This confirms that the message scales with the scaling factor of the receiving node $\mathbf{x}_r$, as required for scale equivariance. Note that in the transformation, we omit permutations $\pi$ as their effect is handled by the inherent design of a Graph Message-passing Network, which is naturally equivariant to node and edge permutations through its aggregation mechanisms.

---

[‡]Let $g_E$ and $g_I$ be equivariant and invariant functions respectively. And let $f(\mathbf{X}) = g_E(\mathbf{X}) \odot g_I(\mathbf{X})$, then $f(q\mathbf{X}) = g_E(q\mathbf{X}) \odot g_I(q\mathbf{X}) = (qg_E(\mathbf{X})) \odot g_I(\mathbf{X}) = q(g_E(\mathbf{X}) \odot g_I(\mathbf{X})) = qf(\mathbf{X})$.

[§]Indeed, $\text{ReScaleEq}(q_1\mathbf{x}_1, \ldots, q_n\mathbf{x}_n) = \bigodot_{i=1}^n \Gamma_i(q_i\mathbf{x}_i)$. Linearity of $\Gamma_i \implies \bigodot_{i=1}^n (q_i\Gamma_i\mathbf{x}_i)$. Element-wise product $\odot \implies (\prod_{j=1}^n q_j)(\bigodot_{k=1}^n \Gamma_k\mathbf{x}_k) = (\prod_{j=1}^n q_j)\text{ReScaleEq}(\mathbf{x}_1, \ldots, \mathbf{x}_n)$.

Finally, having established the scaling properties of the aggregated message $\mathbf{m}$ (derived from the $\text{MSG}_V$ function detailed above), the scale-equivariant update function, $\text{UPD}_V(\mathbf{x}, \mathbf{m})$, is implemented directly as an instance of the ScaleEq function (defined in Equation 7). This direct application is valid because both inputs to the update function (the node's own features $\mathbf{x}$ and the aggregated message $\mathbf{m}$) have been shown to scale by the same factor $q_r$.

### B.4 ScaleInvariant Net

For the scaleGMN as a functional, the network needs to be invariant to CNN weight-space symmetries. This is achieved by applying an invariant readout function that acts on the output of the equivariant network described above. For the readout function $\text{READ}_V$, which maps node features to a graph-level output, Kalogeropoulos et al. [10] employ a strategy that combines canonicalization for scale invariance with Deep Sets [22] for permutation invariance. For hidden nodes, an inner MLP, $\phi$ (which can be integrated with the canonicalization step), transforms each $\tilde{\mathbf{x}}_i$. These transformed hidden features are subsequently aggregated ( via summation: $\sum_i \phi(\tilde{\mathbf{x}}_i)$). Finally, this aggregated representation of hidden nodes, concatenated with features from I/O nodes, is processed by an outer MLP to produce the graph output.

## C  Dataset Details

### C.1 CNN Experiments

For our CNN experiments, we utilize the *Small CNN Zoo* dataset [21]. This dataset comprises 30,000 CNNs, each trained on CIFAR-10-GS [14] and saved at nine distinct training checkpoints. All networks share a standardized 3-layer architecture with 16 filters per layer, resulting in a total of 4,970 learnable parameters per model.

To ensure the feasibility of our analysis in terms of computational cost and to increase the difficulty of the task, we filter networks based on their classification accuracy. Specifically, we retain only those models achieving over 40% accuracy for ReLU-activated networks and over 35% for those using Tanh. This filtering results in two final datasets comprising 10,931 (ReLU) and 12,001 (Tanh) networks. Each dataset is then split into 80% training, 10% validation, and 10% test sets.

The methodology of Unterthiner et al. [21] did not include a validation set, as its primary objective was to gain insights across a diverse sampling of network architectures, not to fine-tune a single model for optimal performance. An inspection of the source code confirms that networks were trained on the complete 50,000-image CIFAR-10-GS training set and evaluated directly on the 10,000-image test set. Our experiments, however, require a validation set. To create one without data leakage, we split the original test set into two equal halves. This results in our final data partition: 50,000 images for training, 5,000 for validation, and 5,000 for testing.

Finally, we pair each CNN training/validation/test split with the corresponding CIFAR-10-GS image data split. This ensures that our ScaleGMN models are evaluated on both network parameters and input images that were not seen during training, preserving the integrity of generalization performance.

### C.2 MLP Experiments

While many model zoos in the literature focus primarily on CNNs, our experimental validation and theoretical exploration also require a corresponding dataset of MLPs. Although Unterthiner et al. [21] include additional MLP experiments in their appendix, they do not release the associated model weights. To the best of our knowledge, the only publicly available MLP dataset is provided by Scabini and Bruno [20]. However, their networks exclude biases and contain significantly more parameters (177,800 trainable parameters), making them unsuitable for our proof-of-concept experiments, which require simpler architectures aligned in scale with our CNN models.

To address this gap, we construct our own *Small MLP Zoo*, designing the data generation process based on the methodologies outlined in the aforementioned works. Each MLP receives as input CIFAR-10-GS images resized to $8 \times 8$ pixels and flattened into a 64-dimensional input vector. All networks in the *Small MLP Zoo* share a uniform architecture: an initial linear layer maps the 64 input features to 48 neurons, followed by two hidden layers with 32 and 16 neurons, respectively, and a

final linear layer outputs 10 neurons for classification. All hidden layers employ Tanh activation functions. The total number of trainable parameters in each network is 5,386.

For each MLP, we randomly sample a configuration (from Unterthiner et al. [21]) consisting of:

- **Optimizer** is chosen uniformly from one of the following: Adam [11], vanilla SGD, and RMSprop [8];
- **Learning rate** is sampled log-uniformly from $[5 \times 10^{-4}, 5 \times 10^{-2}]$;
- $L_2$ **regularization coefficient** is sampled log-uniformly from $[10^{-8}, 10^{-2}]$;
- **Type of weight initializer** is sampled uniformly from Xavier/Glorot normal [4], He normal [7], orthogonal [19], normal, and truncated normal;
- **Variance of weight initializer** is sampled log-uniformly from $[10^{-3}, 0.5]$.

We trained 3,200 unique MLP models on the full CIFAR-10-GS train set, each with a distinct hyperparameter configuration and trained for 15 epochs. To accelerate the process, we utilized parallel training in batches of 40 models, leveraging a vectorized `torch.vmap` implementation tailored for each optimizer. For every model, we saved checkpoints at epochs 5, 10, and 15, capturing the model parameters, optimizer state, and test accuracy. This resulted in a dataset of 9,600 checkpoints.

For our final experiments, we increased task difficulty by filtering the MLPs to include only those with a test accuracy above 0.3. This resulted in a total of 6,135 networks, which we split into 80% training, 10% validation, and 10% test sets. The CIFAR-10-GS data split follows the same methodology used in our CNN experiments.

## D  Hyperparameter Details

*The code, including scripts to generate the results, is presented in `https://github.com/daniuyter/scalegmn_amortization`.*

### D.1  Scale Equivariant Graph Metanetworks

#### D.1.1  Constant Hyperparameters

Our meta-optimizer, $\hat{f}_\phi(G, \boldsymbol{\theta})$, is closely inspired by the ScaleGMN operator architecture introduced in Kalogeropoulos et al. [10] for the INR editing experiment. Specifically, we implement $\hat{f}_\phi(G, \boldsymbol{\theta})$ as a bidirectional GNN, where the GNN follows the traditional message passing paradigm, and includes a residual connection. The parameter update rule is defined as $\boldsymbol{\theta}'_i = \boldsymbol{\theta}_i + \gamma \cdot \text{ScaleGMN}(\boldsymbol{\theta}_i)$, where $\gamma$ is a learnable scaling coefficient (initialized at 0.01), shared across all weights or all biases within the same layer. Within each MLP, we apply layer normalization and use skip connections between successive GNN layers. We fix the hidden dimension for both edge and vertex features to 128. Each MLP in the architecture consists of one hidden layer with a SiLU activation function, and no activation function is applied after the final layer. Additionally, we include node positional encodings but do not use vertex positional encodings.

For the regularization experiments, we fix the $L_1$ coefficient to 0.0015 for the ReLU CNNs, 0.001 for the Tanh CNNs, and 0.0005 for the Tanh MLPs. These $L_1$ values are chosen heuristically to encourage sparsity without incentivizing degenerate solutions, where all weights collapse to zero, which would result in a CE loss of approximately 2.3, equivalent to random guessing on CIFAR-10. We omit ReLU MLPs from our experiments due to unstable training, which we attribute to the computation of reciprocals of edge representations. Similar stability issues are discussed by Kalogeropoulos et al. [10] in Appendix B.2, which addresses the challenges of training positive-scale symmetric bidirectional models.

#### D.1.2  Tuned Hyperparameters

For the remaining hyperparameters, we perform grid searches. Our experiments span multiple dimensions, including model architecture (CNNs and MLPs), regularization (regularized and non-regularized), activation functions (ReLU and Tanh), and whether the scaling symmetry is preserved or broken. Ideally, we would conduct separate hyperparameter sweeps for each of these experimental

settings. However, due to the high computational cost, we limit our analysis to a smaller set of configurations. All grid searches are conducted with a maximum of 300 training epochs, using early stopping if validation loss fails to decrease for 30 consecutive epochs.

For the CNN-ReLU architecture, we perform two separate grid searches, one with and one without $L_1$ regularization, across the following hyperparameters: number of GNN layers {6, 8}, CNN batch size {32, 64}, learning rate {0.0001, 0.0005}, weight decay {0.01, 0.001}, GMN dropout {0, 0.01}, and the percentage of the CIFAR-10-GS train set {0.25, 0.5} used to create $\mathcal{B}$ in Equation (1). The optimal hyperparameters from the grid can be seen in Table 2.

Table 2: CNN-ReLU optimal grid parameters

| $L_1$ | Learning rate | Batch Size | Layers | Weight Decay | Dropout | CIFAR-10-GS Batch |
|---|---|---|---|---|---|---|
| 0 | 0.0005 | 32 | 8 | 0.01 | 0 | 0.5 |
| 0.0015 | 0.0005 | 32 | 8 | 0.01 | 0 | 0.5 |

For the MLP-Tanh architecture, we also perform two separate grid searches, one with and one without $L_1$ regularization, over the following hyperparameters: number of GNN layers {6, 8, 10}, learning rate {0.0001, 0.0005}, weight decay {0.01, 0.001}, and GMN dropout {0, 0.01}. The MLP batch size is fixed at 32, and we use the full train set to construct $\mathcal{B}$ in Equation (1). Empirically, using larger CIFAR-10-GS batches was found to be beneficial for generating a more informative signal for the model to learn from. The optimal hyperparameter settings identified from the grid search are summarized in Table 3.

Table 3: MLP-Tanh optimal grid parameters

| $L_1$ | Learning rate | Batch Size | Layers | Weight Decay | Dropout | CIFAR-10-GS Batch |
|---|---|---|---|---|---|---|
| 0 | 0.0005 | 32 | 8 | 0.001 | 0.01 | 1 |
| 0.0005 | 0.0005 | 32 | 10 | 0.01 | 0 | 1 |

For the CNN-Tanh experiments, we reuse the optimal hyperparameter combinations identified in the CNN-ReLU grid searches. While this approach is suboptimal, it serves as a practical heuristic necessitated by computational constraints. For experiments involving broken symmetry, we use the same configuration as in the corresponding original optimal settings but omit the canonicalization step, thereby breaking scale symmetry while keeping the rest of the architecture unchanged.

## D.2 Baseline Hyperparameters

Performing hyperparameter tuning for the iterative baseline on each network individually is computationally not feasible. To ensure that our hyperparameter selection remains grounded in a quantitative way, we adopt a sampling-based approach. Specifically, we randomly sample 25 networks from the training set and train each for 150 epochs. The hyperparameters yielding the lowest average validation loss across this sample are then selected. The grid search is performed over the following values: learning rate: {0.01, 0.001, 0.0001} and batch size: {64, 128}. The optimal resulting values can be found in Table 4.

Table 4: Optimal hyperparameters for baseline models.

| Model Type | Activation | $L_1$ Coefficient ($\lambda$) | Learning Rate | Batch Size |
|---|---|---|---|---|
| CNN | ReLU | 0 | 0.01 | 64 |
| CNN | ReLU | 0.0015 | 0.01 | 64 |
| CNN | Tanh | 0 | 0.01 | 64 |
| CNN | Tanh | 0.001 | 0.01 | 64 |
| MLP | Tanh | 0 | 0.01 | 64 |
| MLP | Tanh | 0.0005 | 0.01 | 64 |

Note that the learning rates are relatively high, likely due to the small number of training epochs. While training for more epochs is possible, the goal here is not to achieve peak performance indefinitely, but rather to provide a benchmark contextualizing the performance of single-shot fine-tuning, as discussed in Section 3. These hyperparameters are used to fine-tune the full network test sets.

## E   Additional Results

### E.1   ReLU Activation Function CNN Experiment

Table 5 extends the original results (Table 1) for CNNs to with a ReLU activation function. We focus our evaluation on CNNs rather than MLPs, as training with ReLU under bidirectional learning exhibited stability issues in training, particularly for MLPs. These challenges are consistent with earlier observations reported by Kalogeropoulos et al. [10]. Table 5 shows that the ScaleGMN, when used as an amortized fine-tuning model, outperforms simple iterative baseline models, thereby generalizing the results observed for Tanh activations in the main body of the paper.

Table 5: **Evaluation of Amortized Meta-Optimization**. Comparison of CE and $L_1$-regularized experiments on CIFAR-10-GS against iterative baselines for CNN with ReLU activations. Sparsity is defined as the percentage reduction in the total $L_1$ norm of the parameters after fine-tuning. Metanetwork timing was measured as the average forward pass duration over the test set, using a batch size of one. Baseline timing was determined by extrapolating the average per-epoch training time from an initial 30-epoch run, a method found to be consistent across different initializations and activation functions. All measurements were performed on a single NVIDIA A100 GPU.

| Method | Cross-entropy ($\lambda = 0$) | | | L1-regularized ($\lambda > 0$) | | | |
|---|---|---|---|---|---|---|---|
| | Avg Acc (%) | Max Acc (%) | Time (s) | Avg Acc (%) | Test Loss (CE+L1) | Time (s) | Sparsity (%) |
| **CNN Architectures (ReLU)** | | | | | | | |
| *Initial performance* | 44.71 | 55.96 | - | 44.71 | 2.404 | - | 0 |
| *Metanetworks* | | | | | | | |
| ScaleGMN-B | 54.30 | 60.24 | 0.0656 | 42.42 | 1.8442 | 0.0646 | 85.84 |
| GMN-B (sym. broken)[†] | 55.42 | 60.20 | 0.0606 | 41.18 | 1.9512 | 0.0608 | 81.66 |
| *Iterative optimizer* | | | | | | | |
| SGD (25 epochs) | 49.46 | 56.77 | 59.5 | 45.82 | 2.3839 | 82.5 | 49.03 |
| SGD (50 epochs) | 50.29 | 56.10 | 119.0 | 45.94 | 2.1456 | 165.0 | 63.11 |
| SGD (100 epochs) | 51.18 | 56.55 | 238.0 | 45.62 | 1.9862 | 330.0 | 72.24 |
| SGD (150 epochs) | 53.17 | 58.15 | 357.0 | 45.38 | 1.9496 | 495.0 | 75.24 |

[†] Identical to ScaleGMN-B, but without the canonicalization step that ensures scale equivariance.

### E.2   Study of Variability of Results

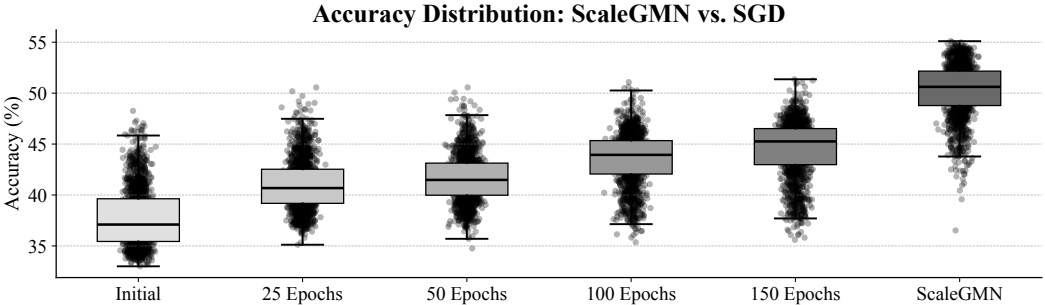

Figure 5: **Accuracy Distribution**. Comparison of the post–fine-tuning accuracy distributions for SGD and ScaleGMN. The boxplots depict results on the CNN-Tanh dataset with $\lambda = 0$.

As shown in Figure 5, our method exhibits comparable variability to traditional iterative optimization while achieving consistently superior accuracy to 150-epoch SGD training with similar interquartile ranges.

The observed variance is theoretically significant, indicating that the network adapts performance based on initial weights $\theta$ to produce optimized parameters $\theta'$. This variance demonstrates genuine parameter transformation rather than fixed mappings, confirming that our approach learns to exploit the structural information in initial weights.

This fundamentally distinguishes our method from weight prediction approaches [12, 23] that generate parameters solely from computational graphs ($\mathcal{G} \rightarrow \boldsymbol{\Theta}$). Our context-aware transformation ($\boldsymbol{\Theta} \times \mathcal{G} \rightarrow \boldsymbol{\Theta}$) leverages initial parameter context to influence optimization trajectories.

