# OpenReview forum: "Symmetry-Aware Fully-Amortized Optimization with Scale Equivariant Graph Metanetworks"
_NeurIPS.cc/2025/Workshop/UniReps — UniReps2025_

### Official Review · Reviewer_VRML · 2025-09-05
**Weak Accept, with some reservations, or weak reject**

**Confidence:** 5

**Review:**

**[Summary]**

This paper proposes an amortized meta-optimizer that performs single‑shot weight updates. For this purpose, the authors apply Scale Equivariant Graph Meta networks. The idea is to map target networks to graphs, and deploy message passing paradigms to respect permutation and scaling symmetries. Empirically, the central positive results are shown on Small CNN/MLP Zoos trained on CIFAR‑10‑GS (with less positive MLP results). The paper also attempts to argue theoretically that CNNs have less scaling gauge freedom than MLPs, which they assert explains why scale‑equivariance helps more on CNNs than MLPs.

**[Strengths]**

- The paper is easy to read, where the construction of the proposed symmetry-aware operator is clear. I appreciate the usage of ScaleGMN in this context.

- There is a nice theoretical story the authors argue to explain their empirical results.

- I like that symmetry-breaking ablations were included.

**[Weaknesses]**

- See below, including an incorrect proof (not a significant deal, as it can be fixed). Also, in the paper's framing, gauge freedom is essentially the same thing as invariance, which is a less fancy word, but principally equivalent here.

- It is not surprising that the symmetry-broken run blows up and fails to complete. In Appendix B.3, the authors deploy a message function that is designed to scale like the receiver $\left(q_r\right)$ under scaling of receiver/sender/edge features by $\left(q_r, q_s, q_e\right)$. The algebra explicitly involves the ratio $q_e q_s^{-1}$. If $q_s$ (sender scale) becomes very small, this ratio explodes. Without canonicalization, such extreme scales clearly are not tamed, which should fatally destabilize training.

**[Incorrect Proof in Core Theoretical Result]**

In Appendix A.3, lines 322-333, the authors argue that because the effective convolution matrix $W$ is doubly‑block Toeplitz with shared entries, any two adjacent rows share at least one kernel weight, so scaling rows independently would break equalities. However, this only holds within an output channel. That is, many rows (different spatial positions) share the same kernel parameters, so those rows must indeed share the same scaling factor. However, this property breaks across different output channels. Each output channel has its own kernel bank, so rows from different channels do not need to have the same scale. The same logic holds for columns grouped by input channel. The current proof only holds for a toy single-channel 2D convolution. This is a mistake in the proof, and can be straightforwardly be fixed by considering the sum of channels across layers (modulo boundary constraints). The conclusion of the lemma is therefore true, but with an incorrect proof.

**[Originality]**

Some prior works exist in this setting (e.g., some are cited in Section 2), which are similar, so I would rate this as moderate. In lines 56-57 in Section 1, the authors claim a significant theoretical contribution (Appendix A.3, Lemma 1) that symmetry gauge freedom is theoretically smaller for CNN than MLPs. However, their proof is wrong. Please see weaknesses above, and this should be fixed. Also, the proof of the Lemma is very straightforward in any case, and I don't think this is a novelty that can be claimed as substantial.

**[Clarity]**

Yes, the paper is clear and easy to read.

- Thanks for the details in Appendix C. I enjoyed reading this section, and it alleviated some of my curiosities.

**[Soundness]**

The experiments seem sound, with some issues mentioned below. The theory, however, needs improvement.

**[Quality of Results]**

Reasonable, in that the paper seems to not go for attaining SoTA, and instead tries to argue for a "proof of concept" approach.

**[Significance]**

I would rate it as borderline, as this work seems to be developing foundations, while (at this stage) brushing some issues under the rug. Example issues being:

- Scalability is a significant issue under this framework. "This meta network is trained on a collection of networks that are trained on a consistent objective"; I find it very difficult to imagine how this technique could be applied on even GPT-2 Small if we're training a neural net for full model weights, not to mention frontier models that singularly require ~300GB to host. As such, the paper evaluates MLP/CNNs. Each CNN has 4,970 parameters per model (Appendix C), which is a far cry from the 8 Billion+ parameter models used in some fields of ML.

- The conversion of a neural network into a graph is also an extreme bottleneck for larger parameter models (if done in the manner described in Appendix B). I consider this technique, in current form, to be completely unusable for ViT-class models.

- The meta network appears to require retraining when a new architecture is specified, or even when there are slight perturbations to the architecture. Is there a notion of transferability/stability under architecture perturbations? This feels quite brittle, which impacts the significance.

- It would have been great if the authors used standard ResNet architectures, which is known to be performant amongst CNN-class nets. I understand that due to needing lots of training data, they used pre-existing (very small scale) models, hence could not use ResNets, but I feel that the paper could be more impactful if larger scale models were used. Especially given that bespoke CNNs may not optimize as well as ResNets. It is unclear if results on models with thousands of parameters generalize to models with.... more parameters, including ones with millions (which come nowhere near SoTA).

**[General Review]**

Not much to comment here. I recommend weak accept (but it is *very* borderline), as it is a fun paper to read.

**[Questions/Suggestions]**

Q1) Lines 56-57, scaling symmetry gauge freedom is smaller for CNN layer than for MLP; would this be a surprising result? Would this not follow from the CNN inherently representing the symmetry structure as a very strong inductive/architectural bias? Similarly, I expect message passing GNNs to have smaller gauge freedom as well, though the paper does not analyze this.

S2) "As fine-tuning networks via a meta network remains unexplored, we do not include other meta fine-tuning methods as baselines". I still suggest that it is a very good idea to include other baselines, not just internal ones, as that confers substantially more validity to the paper's empirical claims.

S3) The paper uses the expression "amortized", which tends to target settings in which case resources are limited (hence why people amortize). However, it seems that the training process for synthesizing the meta network (which is also a neural net) is extraordinarily expensive. This shows up in part in the paper's (very) modest experiments using (very) small models.

- Sure, we can obtain a network that may single shot a "trained" neural network given any initialization of the model parameters if the meta network training goes well, but I find it hard to believe that this is cheaper or more efficient than training a single network properly. I also find it difficult to believe that such single-shotted synthesized model weights perform better than a singular, carefully-tuned model. Why should resources be expended on this meta network than just training a singular (or even multiple) well-trained model(s), given the cost?

- The empirical setup (including the authors' remarks on being bottlenecked by training cost in the Appendix) makes it clear that to train this meta network, one already requires access to an extreme number of already trained model weights, which is simply infeasible for larger neural nets. Beyond the few-thousand parameter models this paper studies, I am unsure how much traction this technique will have.

**[Typos/Minor]**

- [14]/[15] both cite the same work by Kofinas et al.

- [11] and [12] both cite Adam (2014/2017). Sadly, both cite the same arXiv version, with different years (...). I suggest the authors unify the citation, or even better, cite the ICLR version as this paper was published (with an incorrect convergence proof I might add)

- [5] "Goodfellow, Bengio, Courville, and Yoshua Bengio" lists Bengio twice.

- [26] Symmetries, Flat Minima, and the Conserved Quantities of Gradient Flow by Bo Zhao is published at ICLR. To my understanding, Bo Zhao is quite well known in this area of research, and it is best to cite the published version of their work.

- Inconsistent hyphenation of CIFAR‑10‑GS (appears also as CIFAR10‑GS). For instance, lines 488 and 499.

**[Final Recommendation]**

Weak accept or weak reject, the former with some reservations.

**Score:**

3

**Topic Fit:**

3

---

### Official Review · Reviewer_maU2 · 2025-09-13
**Review Comments**

**Confidence:** 2

**Review:**

This paper explores symmetry-aware amortized optimization and proposes a single-shot parameter optimizer, which is more efficient than existing methods that require iterative updates. The method is based on the observation of gauge symmetries in neural network weights, which seems intuitively reasonable. However, I find it difficult to fully grasp the connection between these gauge symmetries and the proposed approach, and I remain somewhat confused about the intuition behind why iterative updates are unnecessary in this framework. That said, the numerical results are promising and appear to support the authors’ claims.

**Score:**

3

**Topic Fit:**

2

---

### Official Review · Reviewer_yup6 · 2025-09-16
**A promising approach to amortized optimization**

**Confidence:** 4

**Review:**

Pros:
- The paper is well-structured and well-written.
- The research problem of efficient network optimization is interesting and significant

Cons:
- The method's reliance on partially trained networks is a practical limitation. Standard training workflows typically run to convergence, especially for the architectures considered in this work. A more compelling setup would involve optimizing networks directly from a random initialization, or optimizing for sparsity on fully-converged architectures.
- The timing comparison is misleading as it does not include the metanetwork's substantial training and hyperparameter tuning costs. The results should include this amortized cost as a separate entry.

Despite the identified weaknesses, which I believe the authors can address when expanding the paper, I recommend for acceptance, cause the topic is worth exploring further.

**Score:**

4

**Topic Fit:**

2

---

### Official Review · Reviewer_eTef · 2025-09-16
**The paper proposes an approach for fully-amortized optimization, enabling single-shot fine-tuning of neural networks. By encoding permutation and scaling symmetries in weight space, the method provides consistent optimization results across equivalent parameterizations. It demonstrates theoretical insights (CNNs have fewer scaling symmetries than MLPs) and empirical gains (fast fine-tuning and sparsification with one forward pass).**

**Confidence:** 3

**Review:**

Strengths:

- The paper shows ScaleGMNs can replace long iterative training with a single forward pass. This enables tasks like fine-tuning or pruning to be done in seconds rather than hundreds of epochs, showing major time and compute savings.
- Experiments show the method can outperform or match SGD run for many epochs. It not only fine-tunes accuracy but also achieves high sparsity (compression) in one step, which is very appealing for efficient deployment.
- Provides a formal proof that scaling symmetry groups differ between architectures (CNNs vs. MLPs), explaining why the method benefits some networks more. This combination of theory + practice strengthens the work.

Weakness:

- The study is restricted to small networks on CIFAR-10. It’s unclear how well the approach scales to large modern architectures (e.g., Transformers) or to other domains like NLP.
- The approach requires a large, diverse “zoo” of networks to train the meta-optimizer which is not ready available at scale for tasks where one would like to leverage it for efficiency.
- Comparisons are mainly against SGD. The paper does not test against other meta-learning or learning-to-optimize baselines, leaving uncertainty about relative competitiveness.

Suggestions:

- Add stronger baseline comparisons with other meta-optimizers or hypernetworks for camera ready version.
- Expansion of theory and application in future work towards transformers since they are ubiquitous to all modern models used in machine learning space.
- An ablation on how many models in zoo are required for efficient training can also be explored in future work.

**Score:**

3

**Topic Fit:**

2